# Reasoning Capabilities of Large Language Models in Dynamic Games of Imperfect Information: A Case Study on Dou Dizhu

## Abstract

The performance of Large Language Models (LLMs) in dynamic games of imperfect information, which demand deep strategic reasoning, remains an underexplored area. This paper investigates this challenge using the popular card game Dou Dizhu as a representative testbed, aiming to enhance the reasoning and decision-making abilities of LLMs in such complex scenarios. First, we establish a rigorous and fair benchmark using a duplicate round-robin tournament to comprehensively evaluate the performance of several state-of-the-art LLMs. This evaluation provided a clear performance baseline and revealed that while these top-tier models are powerful, their significant computational cost motivates the development of smaller, more efficient alternatives. Furthermore, we propose a novel data construction framework designed to bridge the information gap. Its core consists of two unique data curation mechanisms tailored for such games: globally optimal decision alignment via symmetric information and real-time in-game feedback augmentation. By fine-tuning a smaller-scale model on a structured curriculum—comprising this curated data alongside victorious game data—we demonstrate a significant enhancement in gameplay proficiency. The model exhibits a substantially reduced decision error rate and a strategic robustness that significantly outperforms baseline models. Code is available in the Supplementary Material.

## 1 Introduction

A foundational goal of artificial intelligence (AI) is the creation of agents capable of deep reasoning and strategic decision-making in complex environments (Russell et al., 1995; Wooldridge & Jennings, 1995). Historically, the canonical testbed for measuring progress toward this goal has been perfect information games such as Chess and Go. Landmark achievements—from Deep Blue's victory over Garry Kasparov (Campbell et al., 2002) to the superhuman performance of AlphaGo, AlphaZero, and MuZero (Silver et al., 2016; 2017; 2018; Schrittwieser et al., 2020) across various board games—have clearly demonstrated that specialized AI systems, often leveraging deep reinforcement learning (Sutton et al., 1998), can achieve superhuman performance in highly complex strategic domains. However, these successes have been largely confined to environments where all information is transparent to all agents.

A profound paradigm shift has recently emerged in the AI landscape, driven by the rise of general-purpose foundation models, exemplified by LLMs (Brown et al., 2020). Unlike their specialized predecessors, LLMs are not designed for a single task but acquire a wide range of capabilities through large-scale pre-training. Critically, these models have demonstrated remarkable emergent abilities in general-purpose reasoning (Kojima et al., 2022; Wei et al., 2022; Guo et al., 2025), spanning domains from mathematical theorem proving to complex program synthesis (Lewkowycz et al., 2022; Chen et al., 2021). This trend naturally raises a critical question: can this nascent, general-purpose reasoning paradigm be effectively applied to tackle the deep strategic challenges of games, particularly those that lie beyond the perfect information paradigm?

To conduct a rigorous investigation into this frontier, we turn our focus to dynamic games of imperfect information—a class of problems that more closely mirrors real-world strategic uncertainty

and requires agents to reason about the hidden states and intentions of other participants. We select the popular card game Dou Dizhu as our primary case study. As a multi-agent game that integrates both cooperation and competition, Dou Dizhu poses a formidable challenge to an agent's reasoning capabilities, where success relies not only on logical deduction but also on sophisticated opponent modeling (Zha et al., 2021). However, existing work in this area has often treated this as a simple classification or sequence modeling task for fine-tuning (Wang et al., 2025), thereby failing to deeply engage with the core structure of imperfect information games: the absence of information, namely the hidden game states and the intentions and strategies of other players. This motivates us to consider how we can leverage this key challenge to more effectively enhance the reasoning and decision-making abilities of LLMs in such environments.

In this paper, we conduct a systematic empirical investigation into the reasoning capabilities of contemporary reasoning-focused LLMs in this challenging environment. We first establish a rigorous and fair benchmark using a duplicate round-robin tournament, a format designed to neutralize the stochasticity of card distribution. We employ this format to comprehensively benchmark several leading LLMs, which provided us with a clear performance baseline quantifying the capabilities of state-of-the-art models on this complex task. This benchmark inspired our core research motivation: can we, through a more efficient, data-driven approach, significantly elevate the strategic reasoning of a smaller model to approach the performance of these top-tier LLMs? To achieve this goal, we propose a novel data construction framework designed to bridge this information gap. The core of this framework lies in two unique data

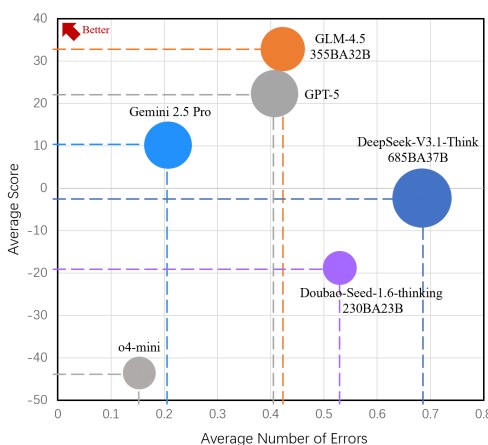

Figure 1: Performance comparison of frontier LLMs on the Dou Dizhu benchmark.

curation mechanisms tailored for the challenges of imperfect information games: 1) Globally Optimal Decision Alignment via Symmetric Information: We introduce a pioneering "post-hoc validation" mechanism. After a model makes a decision under imperfect information, we reveal the hidden hands of all players to it (creating a temporary "God's-eye view") and prompt it for re-evaluation. 2) Real-time In-Game Feedback Augmentation: In a multi-agent simulation, virtual opponents and teammates provide strategic evaluations for each move the model makes.

By meticulously constructing a structured curriculum from these two types of rigorously curated data, combined with victorious game data, we fine-tune a smaller-scale reasoning model. Experimental results demonstrate that this framework significantly enhances the model's gameplay proficiency, leading to a substantially reduced decision error rate and a strategic robustness that far surpasses its baseline counterpart. The primary contributions of this work are threefold:

- We establish the first fair and reproducible benchmark for evaluating the strategic reasoning of reasoning-focused LLMs in the complex, multi-agent, imperfect information game of Dou Dizhu.

- We introduce a novel data construction framework that leverages simulated feedback and counterfactual validation to generate high-quality training data, offering a new and efficient pathway for enhancing the strategic reasoning of LLMs.

- We provide empirical evidence that a data-centric approach can effectively instill expert-level reasoning into smaller models, presenting a viable direction for developing cost-effective, efficient, and powerful AI agents for complex strategic domains.

## 2 RELATED WORKS

Research on artificial intelligence for imperfect information games has historically been dominated by specialized, computationally intensive algorithms. Seminal works like DeepStack (Moravčík

et al., 2017), Libratus (Brown & Sandholm, 2018) and DouZero (Zha et al., 2021) achieved super-human performance in poker games, through deep reinforcement learning and extensive self-play. While powerful, these approaches create highly specific, opaque models that are difficult to generalize.

The recent rise of LLMs has introduced two new paradigms. The first paradigm utilizes LLMs as zero-shot reasoners, leveraging sophisticated prompt engineering techniques such as Theory of Mind (ToM) (Guo et al., 2023; Yim et al., 2024) or recursive thinking (Duan et al., 2024) to elicit strategic behavior without any model training. These methods are flexible but are fundamentally limited by the inherent capabilities of the pre-trained LLMs.

The second, more prevalent paradigm focuses on adapting LLMs through supervised fine-tuning (SFT). Studies like PokerGPT (Huang et al., 2024) and PokerBench (Zhuang et al., 2025) have successfully demonstrated that fine-tuning on high-quality game data can significantly enhance an LLM's performance in specific games. These works correctly identify that the quality of the training data is paramount to success. However, their methodologies for improving data quality primarily revolve around filtering and selecting from pre-existing datasets of expert play. We argue that this approach, while effective, is passive and does not fully address the core challenge of imperfect information games: reasoning under uncertainty. A crucial gap remains in developing methods to more actively and explicitly instill the principles of strategic reasoning into the training data itself.

## 3 METHOD

In this section, we introduce the comprehensive methodology designed to evaluate and enhance the strategic reasoning of LLMs in Dou Dizhu. First, in Section 3.1, we detail the Dou Dizhu benchmark, a rigorous tournament framework we developed for fair and robust model evaluation. To provide essential context, we briefly summarize the game rules of Dou Dizhu in Section 3.2. Subsequently, in Section 3.3, we present our core contribution: a novel framework for the construction of data, which leverages two unique mechanisms to create a high-quality dataset. Finally, in Section 3.4, we describe our curriculum learning strategy, which organizes this data to progressively fine-tune a smaller model and elevate its gameplay proficiency.

### 3.1 DOU DIZHU BENCHMARK

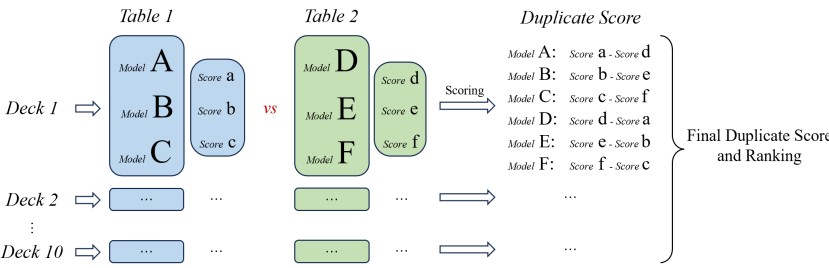

Figure 2: The Architecture of our duplicate round-robin tournament benchmark. The benchmark ensures fairness by creating paired tables where players in corresponding positions are dealt identical hands from duplicate decks. Player seating arrangements are systematically rotated to cover all combinatorial possibilities.

**Motivation and Design Philosophy**   Evaluating the true strategic capabilities of an agent in card games like Dou Dizhu is notoriously challenging due to the game's inherent stochasticity. The random distribution of cards in any given hand can significantly confound performance metrics, making it difficult to disentangle genuine skill from luck. To overcome this fundamental challenge and enable a fair and rigorous comparison of LLM agents, we introduce the Dou Dizhu benchmark. Its design philosophy is centered on isolating strategic skill as the sole determinant of performance.

**Core Mechanism: The Duplicate Format** The cornerstone of our benchmark is the adoption of a duplicate tournament format, a methodology inspired by competitive bridge to neutralize the element of chance. For each pre-generated "deal" (a specific distribution of the 54 cards among the three players and the bottom cards), we create multiple, simultaneous games or "tables." The crucial principle is that agents occupying the same relative position across different tables are dealt the exact same initial hand. This setup ensures that any variation in the outcome of the deal is attributable solely to the strategic decisions made by the agents, rather than the quality of the cards they received.

**Tournament Structure: Round-Robin Rotation** To ensure a comprehensive and unbiased evaluation, the benchmark employs a round-robin structure. Each LLM agent is systematically rotated through every possible seating arrangement and against every other opponent agent. For a set of $N$ agents, we ensure that each agent plays a statistically significant number of deals from each of the three positions. This systematic rotation guarantees that the final performance metrics are not biased by a particular set of opponents or a favorable seating position, thereby covering all relevant combinatorial matchups.

**Evaluation Metrics** We utilize two distinct metrics to measure performance within the benchmark. The first is the score, we will detail in Section 3.2. The second, and more critical, metric is the duplicate score. For any given deal and player position, an agent's duplicate score is calculated by comparing its raw score against the score of all other agents who played the exact same hand in the same position. A positive duplicate score signifies an above-average strategic performance for that particular deal. The final ranking of the agents is determined by their average duplicate score across all deals, providing a robust, skill-based measure of their true capabilities.

In summary, the Dou Dizhu benchmark provides a robust and reproducible framework for evaluating agent performance by neutralizing stochastic elements. This allows for a direct and fair comparison of the underlying strategic reasoning and decision-making abilities of LLM agents.

## 3.2 Game Rules of Dou Dizhu

To ensure the standardization and authority of our experimental environment, all games in this study strictly adhere to the official competition standards promulgated by the Chess and Card Sports Management Center of the General Administration of Sport of China, as detailed in the Official Rules for Competitive Dou Dizhu (Two-vs-One Poker Game) (ISBN: 978-7-5009-4953-4) [1].

Distinct from studies that may utilize simplified environments, our implementation of the game flow is complete and comprehensive. Each game encompasses all official phases, including Bidding, Determining the landlord, Doubling, Redoubling, Card Playing, and final Scoring. Throughout this entire process, all decisions are made autonomously by the LLM agents without any human intervention. The detailed in-game scoring methodology also follows the official rules, the specifics of which are provided in Section A.2.

To support our large-scale experiments, we developed a fully-featured game interaction framework. This framework automates the interactions for the LLM agents and dynamically generates the corresponding prompts based on the game state. This framework was consistently applied across both our benchmark evaluation and the subsequent data construction phases. For implementation details of the framework and the prompt construction mechanism, please refer to our Supplementary Material and Section A.3.

## 3.3 Construction of data

To efficiently instill expert-level strategic reasoning into a smaller model, we eschewed the conventional reinforcement learning paradigm, which relies on extensive game experience, in favor of a data-centric construction framework. Our core philosophy is that the quality, not merely the quantity, of training data is the key determinant of a model's upper capability limit.

---

[1]https://www.sport.gov.cn/qpzx/n5384/c726699/part/412005.doc

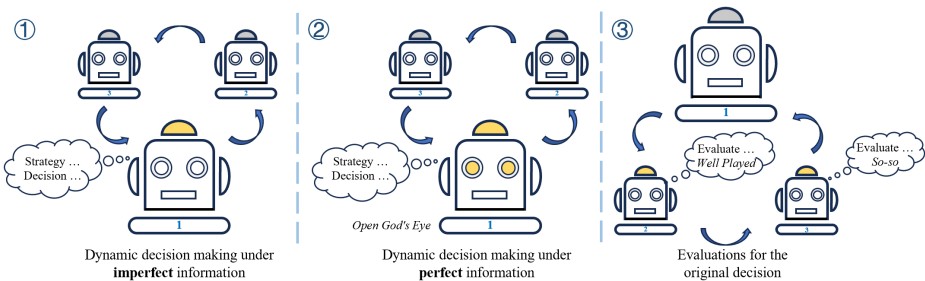

Figure 3: Data construction: Standard data, globally optimal decisions under perfect information, and real-time decision feedback.

### 3.3.1 FOUNDATIONAL DATASET: DISTILLATION AND VICTORIOUS DATA

We begin by constructing a high-quality foundational dataset, which provides a strong starting point for the training of our smaller model. The construction of this dataset consists of two processes:

**Expert Knowledge Distillation** We selected leading open-source Large Language Models (DeepSeek R1 and GLM-4.5) renowned for their powerful general reasoning capabilities to serve as "expert teachers." We had these expert models engage in a large number of self-play games within our custom-built game framework, recording their complete decision-making processes, including their analysis of the game state and their final action choices (Xu et al., 2024). Statistical details are provided in Table 1. This data forms the main body of our distilled dataset.

Table 1: Statistics of the distilled data from expert teacher models. The "Avg. Decisions" refers to the number of times the model make an actual card-playing choice under imperfect information.

| Teacher Model | # Games | # Interactions | Avg. Decisions | Avg. Output Tokens |
|---|---|---|---|---|
| DeepSeek R1 | 673 | 71,648 | 35.82 | 2,709.89 |
| GLM-4.5 | 5,448 | 665,347 | 41.04 | 705.86 |

**Victorious Game Filtering** Building upon the distilled data, we performed an initial, simple filtering pass. We retained only the complete action trajectories of the players who ultimately won the game. The underlying assumption is that a winner's decision sequence is, on the whole, more likely to contain superior strategies.

### 3.3.2 DATA SCREENING MECHANISM FOR IMPERFECT INFORMATION

We posit that relying solely on victory filtering is insufficient, as the resulting data still contains a substantial amount of flawed reasoning and decisions made under the constraints of limited information. To address this, we designed two unique data curation mechanisms specifically tailored for the characteristics of imperfect information games (Figure 3).

**Globally Optimal Decision Alignment via Symmetric Information** The core of this mechanism is the introduction of a "post-hoc validation" process to assess the robustness of a decision from a "God's-eye view." For any decision made under normal, imperfect information conditions, we conduct a counterfactual test: we reveal the hidden hands of all players to the model and prompt it to re-evaluate its choice with this complete, symmetric information. Only when the model's decision remains consistent across both informational conditions do we label the original decision as a high-quality "golden sample." Such a decision is not only justifiable from the player's limited perspective at the time but also holds up under the scrutiny of global optimality, having successfully overcome the interference of the information gap.

**Real-time In-Game Feedback Augmentation** Beyond the optimality of one's own decision, reasoning about the intentions of other players and their evaluation of that decision is a critical aspect of imperfect information gameplay. We therefore introduced a feedback-based curation mechanism. Within our multi-agent simulation environment, for every move a model makes, we prompt the virtual Opponent and Teammate to provide a strategic evaluation of that move. This evaluation is based on their own hands and game objectives. For instance, a move that helps a teammate successfully play out their remaining small cards would receive positive feedback from the teammate, while a move that easily cedes the turn to an opponent would receive negative feedback. We quantify these evaluations into a score (from -5 to 5) and filter for high-scoring decisions. Through this method, the data we select is not only effective from a self-centric perspective but is also proven to be potent on an inter-agent, game-theoretic level.

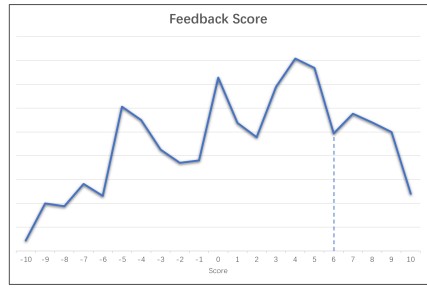

Figure 4: The distribution of feedback scores. The scores from both teammate and opponent are combined.

We integrated both mechanisms to curate our final high-quality dataset. Statistical analysis revealed that the proportion of decisions that achieved alignment with the globally optimal choice was 76.26%. The distribution of the feedback scores is presented in Figure 4. We set a threshold of 6 (out of 10), selecting only the most optimal decisions that also received a feedback score of 6 or greater, resulting in a final dataset of 41,884 samples.

### 3.4 CURRICULUM LEARNING

Based on the datasets constructed in the previous section, we designed a structured curriculum learning strategy (Bengio et al., 2009). This approach, which advocates for learning from easier to harder examples, has recently been demonstrated to be highly effective for fine-tuning LLMs as well (Kim & Lee, 2024; Liu et al., 2024). Our curriculum is divided into two stages, with the corresponding data volumes detailed in Table 2.

**Winning Strategy Learning** First, we fine-tune the model on the victorious trajectory data. The objective of this stage is twofold: to allow the model to learn the fundamental rules and common playing patterns of Dou Dizhu, while simultaneously enabling it, on this low-noise dataset, to focus on the more effective strategic patterns that lead to victory. We employ a low learning rate and utilize an early stopping mechanism to prevent overfitting.

Table 2: Data distribution. Our curriculum learning uses Victorious data and GOFA data.

| Total | Victorious | GOFA |
|---|---|---|
| 247,705 | 85,079 | 41,884 |

**Expert-Level Reasoning Alignment** Then, we conduct fine-tuning using our highest-quality GOFA (Globally Optimal & Feedback-Augmented) dataset. Every sample in this dataset has undergone dual validation, representing the most strategically robust reasoning chains and the most optimal decisions.

Throughout the entire training process, we utilize QLoRA (Quantized Low-Rank Adaptation) (Dettmers et al., 2023) for parameter-efficient fine-tuning. A noteworthy aspect of our methodology is the validation mechanism, which is not based on a traditional loss function. We hold out a portion of the GOFA data as an independent validation set. During each validation step, we compute the model's decision accuracy on this set and use it as the sole metric for model selection and early stopping. We argue that, compared to cross-entropy loss, which can be influenced by token output probabilities, decision accuracy serves as a more direct and reliable measure of a model's true performance in strategic gaming tasks.

## 4 EXPERIMENTS

In this section, we systematically evaluate our proposed method through a series of empirical studies. First, we test the performance of current state-of-the-art (SOTA) LLMs on our constructed Dou Dizhu benchmark. Second, we evaluate, through comparative experiments, the extent of the gameplay proficiency improvement for a smaller model, brought by our proposed data-centric construction framework and curriculum learning strategy. Finally, we want to know, within our framework, how much of a performance contribution the GOFA method can provide for imperfect information games.

### 4.1 EXPERIMENTAL SETUP

#### 4.1.1 MODEL SELECTION

**SOTA Model** To test the level that existing reasoning models can achieve in the field of imperfect information dynamic games and to obtain their rankings, we selected several current industry-leading closed-source and open-source LLMs for competition, including GPT-5, Gemini 2.5 pro, GLM-4.5, o4-mini, DeepSeek-V3.1-Think and Doubao-Seed-1.6-thinking. We use them by calling their APIs.

**Comparative Model** We select Qwen3-4B-Thinking-2507 as the target "student model" for our method. To facilitate quantification, we selected a subset of reasoning models from the Qwen family (Yang et al., 2025) as comparative objects, including Qwen3-Next-80B-A3B-Thinking, Qwen3-30B-A3B-Thinking-2507, Qwen3-14B, and Qwen3-8B.

#### 4.1.2 TRAINING DETAILS

All fine-tuning was conducted using the QLoRA. We set rank to 16, alpha to 32, and dropout to 0.05. The Paged AdamW optimizer was used, with learning rates set to 3e-5 and 1e-4 during different stages of the curriculum. We randomly partitioned 1,000 samples as the validation set. The early stopping was based on the decision accuracy on this validation set. All training and inference were performed on RTX 5090 GPUs.

### 4.2 BENCHMARK RESULTS OF SOTA LLMS

The competitive results and rankings of the SOTA LLMs on the Dou Dizhu benchmark are presented in Figure 1 and Table 3. We conducted 20 complete duplicate round-robin tournaments, comprising a total of 200 unique deals and 400 matches.

Table 3: Performance of SOTA LLMs on the Dou Dizhu benchmark. Here, an "Error" indicates a decision that violates the game rules. The error count is averaged per match. Models are ranked by their average duplicate score.

| Model | Avg. Duplicate Score | Avg. Errors | Avg. Output Tokens |
|---|---|---|---|
| GLM-4.5 | **32.75** | 0.43 | **571.74** |
| GPT-5 | 22.20 | 0.41 | 1424.99 |
| Gemini 2.5 Pro | 10.05 | 0.21 | 1839.10 |
| DeepSeek-V3.1-Think | $-2.45$ | 0.69 | 2241.21 |
| Doubao-Seed-1.6-thinking | $-18.90$ | 0.53 | 1736.99 |
| o4-mini | $-43.65$ | **0.16** | 1095.55 |

The evaluation results in Table 3 lead to several key observations. First, GLM-4.5 demonstrates the strongest overall gameplay capability, with an average duplicate score of 32.75, significantly outperforming the other models. This underscores its powerful potential in complex strategic reasoning tasks. It is followed by GPT-5 and Gemini 2.5 Pro, which also achieved positive duplicate scores, indicating robust decision-making.

Second, an interesting trade-off between performance and efficiency emerges. GLM-4.5 not only leads in score, but its average output token count is also substantially lower than other high-

performing models, exhibiting exceptional decision efficiency—a result that was unexpected. Conversely, some models, such as DeepSeek-V3.1-Think, generated a large volume of text during their reasoning process, which did not translate into effective strategies and, in fact, led to a higher rate of decision errors.

Finally, we observe that the number of decision errors does not have a simple linear relationship with the final score. For instance, o4-mini has the lowest average error count among all models, yet it achieved the lowest duplicate score. This suggests that in certain scenarios, a model's reasoning ability and its instruction-following capabilities may be in conflict.

## 4.3 COMPARATIVE EXPERIMENTS

To comprehensively evaluate the effectiveness of our proposed data construction framework and curriculum learning strategy, we conducted a series of rigorous comparative experiments. The Qwen3-4B-Thinking-2507 model was trained for approximately one epoch in each of the two curriculum stages. The model checkpoint that achieved the highest decision accuracy on the validation set was selected for the final evaluation. After training, we directly compared our model against other reasoning models from the Qwen family on the Dou Dizhu benchmark, again conducting 20 complete duplicate round-robin tournaments. The results are presented in Table 4.

Table 4: Performance comparison of our fine-tuned model against other models from the Qwen family.

| Model | Avg. Duplicate Score | Avg. Errors | Avg. Output Tokens |
|---|---|---|---|
| Qwen3-Next-80B-A3B-Thinking | **61.40** | 0.80 | 5155.61 |
| Qwen3-30B-A3B-Thinking-2507 | 44.35 | 0.66 | 3146.71 |
| **Qwen3-4B-GOFA (Ours)** | 17.25 | **0.33** | **863.52** |
| Qwen3-8B | $-22.35$ | 0.51 | 1177.44 |
| Qwen3-14B | $-34.85$ | 0.43 | 1012.06 |
| Qwen3-4B-Thinking-2507 | $-65.80$ | 1.06 | 1145.74 |

Notably, the Qwen3-4B-GOFA model, produced by our methodology, demonstrates a tremendous improvement over its baseline, surpassing its larger counterparts in the same model series. Its average duplicate score surged to +17.25 from the baseline's -65.80. Furthermore, its average number of decision errors per match is the lowest among all compared models, and its average output token count also decreased, showcasing enhanced robustness and superior reasoning efficiency. This provides strong evidence that our method not only strengthens the model's reasoning capabilities in dynamic games of imperfect information but also improves its instruction-following ability and decision-making reliability.

In summary, our Qwen3-4B-GOFA model exhibits exceptional improvements across the three dimensions of performance, efficiency, and reliability. This validates that our data-centric framework is an effective and viable pathway for empowering smaller models to achieve expert-level proficiency in complex strategic games.

## 4.4 ABLATION STUDY

To quantify the contribution of our GOFA methodology, we designed a rigorous ablation study. We primarily compared the following three models: Qwen3-4B-Thinking-2507 (the base model), Qwen3-4B-Victorious (a model fine-tuned only on victorious data), and Qwen3-4B-GOFA (our final model).

For comparison purposes, we designed a special triangular duplicate tournament. In each round of this tournament, the three models were matched against each other three times with seat rotations, ensuring that each hand was played by every model. We again conducted 20 rounds of these triangular duels, comprising a total of 60 deals and 60 matches. Different from the previous setup, each duplicate duel here is equivalent to three deals played on the same set of dealt cards.

The results of the ablation study, presented in Table 5, clearly and forcefully demonstrate the step-by-step efficacy of our data construction framework and curriculum learning strategy.

Table 5: Ablation study results of our data construction framework.

| Model | Avg. Duplicate Score | Avg. Errors | Avg. Output Tokens |
|---|---|---|---|
| **Qwen3-4B-GOFA** | **20.35** | **0.36** | 823.76 |
| Qwen3-4B-Victorious | 9.40 | 0.53 | **810.44** |
| Qwen3-4B-Thinking-2507 | $-29.75$ | 0.98 | 1100.23 |

Fine-tuning on only the victorious data in the first stage already brings a substantial performance improvement. The model's average duplicate score increases dramatically, and its error rate is nearly halved. This proves that a degree of imitation learning is effective for enhancing not only the model's instruction-following ability but also its reasoning and decision-making capabilities. The most critical finding is that our final model, after being fine-tuned on our GOFA data, achieves another qualitative leap in performance. This result provides a reliable validation that our two core data curation mechanisms—globally optimal alignment and real-time feedback augmentation—are a robust means for enhancing the model's deep reasoning ability and decision-making reliability.

## 5 CONCLUSION

In this work, we systematically investigated the strategic reasoning capabilities of LLMs in the complex, imperfect information game of Dou Dizhu. We first established a fair and reproducible benchmark using a duplicate round-robin tournament to provide a clear quantitative assessment of SOTA LLMs. Our primary contribution is a novel, data-centric framework that constructs a high-quality dataset through two innovative mechanisms: globally optimal decision alignment and real-time in-game feedback augmentation. By fine-tuning a 4B-parameter model on a structured curriculum using this data, we demonstrated a significant leap in performance, reliability, and efficiency. Our findings validate that a data-centric approach, centered on high-quality data engineering, is a viable and efficient pathway for empowering smaller models with expert-level reasoning in complex strategic domains.

## 6 LIMITATIONS AND FUTURE WORK

While our method achieved encouraging results, this study is subject to several limitations that also open avenues for future work.

**Static Knowledge and Dynamic Game Evolution** Our framework relies on knowledge distilled from static teacher models. Although the gameplay itself is dynamic, our agent's core knowledge base is not acquired through a process of continuous self-evolution. A key direction for future work is to explore the integration of our data-driven approach with self-play mechanisms to create an agent capable of perpetual self-improvement.

**Boundaries of Adversarial Robustness** It is important to note that our data construction framework, particularly the real-time feedback mechanism, largely addresses the challenge for traditional models in handling deceptive strategies such as bluffing. However, the boundaries of this robustness are defined by the strategic diversity of our teacher models. Our model is highly optimized to compete against rational or semi-rational "LLM-style" opponents. Its performance against the broader, and at times deeply irrational, spectrum of strategies exhibited by human players remains to be further validated. Future work could explore introducing human-in-the-loop data or specially trained adversarial agents to further expand the model's strategic adaptability.

**Generalization to Broader Game Types** All experiments in this study were focused on the specific game of Dou Dizhu. We believe the core concepts of our data construction framework possess strong generalizability. However, due to computational resource constraints, we were unable to extend the framework to other types of imperfect information games (e.g., Poker, Bridge) for validation. Future work will aim to test the generalizability of our methodology in a wider range of strategic environments.

## 7 ETHICS STATEMENT

The research presented in this paper strictly adheres to the ICLR Code of Ethics. Our work is centered on the study of strategic reasoning in the context of the card game Dou Dizhu, a simulated environment. The data used for our experiments was generated through self-play between publicly available LLMs, and no human subjects or personally identifiable information were involved in any stage of our research. The datasets constructed and used in this study do not contain any sensitive or private content.

We believe our research contributes positively to the scientific understanding of the capabilities and limitations of LLMs in complex, multi-agent strategic domains. The primary goal of this work is to advance knowledge in the field of artificial intelligence. We do not foresee any direct negative societal impacts or potential for misuse arising from our methodology or findings.

## 8 REPRODUCIBILITY STATEMENT

To ensure the reproducibility of our research, we have provided a comprehensive account of our methodology and experimental setup. All key components required to replicate our findings are detailed across the main paper, the appendix, and the supplementary material.

**Benchmark and Game Environment**   The architecture of our Dou Dizhu benchmark, including the duplicate round-robin tournament protocol, is described in Section 3.1. The specific game rules followed are detailed in Section 3.2, with full scoring mechanics provided in Section A.2.

**Data Construction and Curation**   Our novel data construction framework is presented in Section 3.3. This includes a detailed description of our two core data curation mechanisms: globally optimal decision alignment and real-time in-game feedback augmentation. Statistics of the generated datasets are provided in the main text.

**Training and Evaluation**   The curriculum learning strategy is outlined in Section Section 3.4. Key hyperparameters, the model architecture, and the specifics of the training setup are detailed in Section 4.1. Our evaluation metrics, including the Duplicate Score, are formally defined in Section 3.1.

**Code and Prompts**   The complete source code for our game interaction framework, data processing scripts, training procedures, and evaluation protocols is available in the Supplementary Material. The supplementary material also includes the exact prompt templates used to interact with the LLM agents, as detailed in Section A.3.

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

## A APPENDIX

### A.1 THE USE OF LARGE LANGUAGE MODELS

Large language models were used as research and writing assistants during the preparation of this manuscript. Their use included (1) finding and summarizing related work to help situate our research within the current literature; (2) refining sentence structure, correcting grammar, and improving clarity; and (3) translating technical descriptions and suggesting appropriate academic terminology. All core ideas, experimental designs, and conclusions are the original work of the authors, who thoroughly reviewed and edited the final text for accuracy and integrity.

### A.2 DETAILED SCORING RULES OF DOU DIZHU

The score for each defender is calculated as: the landlord's bid value × the win/loss coefficient × the doubling multiplier. The landlord's score is the negative sum of the scores of the two defenders.

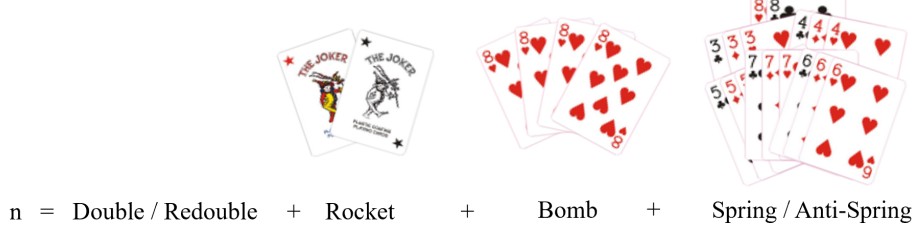

n = Double / Redouble + Rocket + Bomb + Spring / Anti-Spring

Figure 5: Calculation method of the exponent of the doubling factor.

**Win/Loss Coefficient** The win/loss coefficient is set to +1 if the defenders win the game, and -1 if they lose.

**Doubling Multiplier**    The doubling multiplier is calculated as $2^n$, where the exponent n is the sum of the counts of the following in-game events (also shown in Figure 5):

- The number of Rockets played (a pair of Jokers).
- The number of Bombs played (four cards of the same rank).
- A value of 1 for a "Spring" victory (the landlord wins without the defenders playing any cards).
- A value of 1 for an "Anti-Spring" victory (the defenders win without the landlord playing more than their first turn).
- A value of 1 for each defender who chose to "Double" during the bidding phase.
- A value of 1 if the landlord chose to "Redouble" during the bidding phase.

**Asymmetrical Scoring for Defenders**    The scores for the two defenders are calculated individually, as the doubling multipliers from the bidding phase may affect them differently.

- Defender's "Double": If a defender chooses to "Double," that specific multiplier (typically x2) applies only to that defender's individual score calculation. The other defender's score, if they did not double, is not affected by this specific multiplier.
- Landlord's "Redouble": In contrast, if the landlord chooses to "Redouble," that multiplier (typically an additional x2 factor) is a global effect that applies to the score calculations of both defenders, regardless of their individual doubling choices.

### A.3    PROMPT CONSTRUCTION AND TEMPLATES

The invocation of our LLM agent for any given decision point—whether during training or inference—is facilitated by the programmatic construction of a textual input prompt. This prompt is dynamically tailored to the specific game state and the required action, ensuring that the LLM can make an optimal decision based on rich contextual information. Table 6 provides a structural overview of this prompt construction process, detailing the key informational components provided to the agent for each type of decision.

Table 6: Structural overview of the programmatically generated prompts. Each prompt consists of common contextual components and a task-specific module.

| Prompt Component | Objective | Key Information Provided |
|---|---|---|
| *Common Components (Included in all prompts)* | | |
| **System Prompt** | Set the foundational game rules. | A comprehensive description of the Dou Dizhu rules, player objectives, and scoring principles. |
| **Base Context** | Establish the player's role and current situation. | Player name, role (landlord/defender), full game history, and the player's current hand. |
| *Action-Specific Modules* | | |
| **Bidding** | Prompt the agent to bid for the landlord position. | Bidding rules and the valid point range (0-3). |
| **Doubling** | Prompt the agent to decide whether to double the stakes. | Information on whether other players have already doubled. |
| **Playing** | Prompt the agent to play a valid combination of cards or pass. | Remaining card counts of others, the last valid hand played, the number of consecutive passes, and a partial list of playable combinations. |
| **Feedback** | (For Data Curation) Prompt to evaluate previous moves made by others. | The last one move made by other players, their roles relative to the agent (ally/opponent), and a detailed [-5, 5] scoring rubric. |