# OpenReview forum: "Reasoning Capabilities of Large Language Models in Dynamic Games of Imperfect Information: A Case Study on Dou Dizhu"
_ICLR.cc/2026/Conference — Submitted to ICLR 2026_

### Official Review · Reviewer_fnX9 · 2025-10-19

**Soundness:** 2
**Presentation:** 1
**Contribution:** 2
**Rating:** 2
**Confidence:** 4

**Summary:**

This paper studies LLM strategic reasoning in a dynamic imperfect-information game (Dou Dizhu). The authors implement a duplicate round-robin tournament framework with a "duplicate score" metric (which directly reflects how good players are). They propose a data-centric training recipe for a small student model that was distilled from DeepSeek R1 and GLM-4.5 internal gaming records with the "Globally Optimal and Feedback-Augmented" (GOFA) strategies: (1) Globally Optimal: post-hoc symmetric-information re-evaluation ("God’s-eye view") and (2) Feedback-Augmented: real-time teammate/opponent feedback. Their results show that the small distilled model (Qwen3‐4B‐GOFA) exhibits exceptional improvements across the three dimensions of performance, efficiency, and reliability against Qwen counterparts.

**Strengths:**

1. The metric design is controllable: the duplicate format and round-robin seating reduce variance and position bias. The final "duplicate score" metric across all deals reflects better decision-making rather than luck.
2. They include an ablation comparison with the triangular duel, showing continuous gains and error reductions (Base → Victorious → GOFA).
3. The duplicate score for the 4B student model improves significantly with fewer errors and more efficient outputs, providing a low-cost training pipeline to improve small models in gaming environments compared to RL or self-play.

**Weaknesses:**

1. **Unclear and Unsubstantiated Evaluation Metrics.**
The definition and presentation of the evaluation metrics are unclear, high-level, and conceptual. Section 3.1 (Evaluation Metrics) introduces "Score" as the first metric, but this metric is never used or reported in the experimental results. All quantitative analyses in Section 4 rely solely on “Average Duplicate Score”, which itself is never formally defined.
For the "Duplicate Score", although the authors briefly describe a “duplicate-round” setup and provide Figure 2 as an illustration, neither the figure nor the metric is mathematically specified or explained. The absence of a clear formula and the illustration of figures make the evaluation ambiguous, weakening the empirical rigor of the entire study.


2. **Missing Strong Baselines and Cross-Section Comparisons.**
The paper claims the first fair benchmark but does not compare to strong Dou Dizhu agents from RL (e.g., DouZero[1]). Furthermore, Section 4.1 evaluates teacher models against existing SOTA models, but Section 4.2 evaluates only the student model against other Qwn counterparts, with no overlap or cross-comparison between the two groups. Therefore, I cannot determine where the trained models stand relative to existing SOTA models. The current results only show that the student model surpasses some of its Qwen counterparts, but its absolute performance level is unclear.


3. **Distribution-Dependent and Non-Comparable Evaluation Metric.**
The “Duplicate Score” metric has a fundamental limitation: It measures only relative performance rather than absolute capability. This score depends on the strength distribution of the opponents, so the values are not globally comparable across different model sets. Therefore, I think this metric (together with weakness 2) cannot support claims of “expert-level parity”. For benchmarking, it needs a more meaningful and reproducible evaluation metric. For example, the author can use a fixed standardized baseline model and evaluate all other models against it. Or simply include all pairwise matchups among evaluated models, but this is unscalable and computationally expensive.


[1] DouZero: Mastering DouDizhu with Self-Play Deep Reinforcement Learning. Zha et al.

**Questions:**

See weaknesses.

1. Could you provide a formal mathematical definition of the Average Duplicate Score metrics?

2. How do you ensure comparability of Duplicate Scores across tournaments with different opponent sets?

3. Can you provide results comparing Qwen3-4B-GOFA directly against teacher or SOTA baselines?

4. Have you compared your models against RL-based Dou Dizhu agents such as DouZero or human expert data?

5. Could you elaborate on the GOFA filtering process?

---

### Official Review · Reviewer_eiBc · 2025-10-27

**Soundness:** 2
**Presentation:** 2
**Contribution:** 2
**Rating:** 2
**Confidence:** 5

**Summary:**

This paper aims to enhance the strategic reasoning capabilities of Large Language Models (LLMs) in the Dou Dizhu scenario. Overall, it constructs rollout data using expert LLMs, filters the data through two mechanisms: globally optimal decision alignment via symmetric information and real-time in-game feedback augmentation, and finally fine-tunes the LLM with this curated data to improve its performance.

**Strengths:**

The proposed method demonstrates certain effectiveness.

**Weaknesses:**

* Constructing game environment rollout data with expert models and then fine-tuning smaller LLMs to boost performance is a fairly common practice, lacking innovation.
* Even though the paper proposes methods for filtering data and improving data quality, the overall innovation is still low. Approaches like curriculum learning are also well-established, and no uniqueness tailored to the Dou Dizhu scenario is observed.
* While the paper indeed conducts comprehensive work around data construction, its innovation falls far short of the requirements for ICLR.

**Questions:**

1. The benchmark evaluation uses only 200 unique deals across 20 tournaments, which appears insufficient for a high-variance card game to establish statistically robust conclusions. The paper provides no confidence intervals, p-values, or statistical significance tests to validate that the observed performance differences are not due to random chance. Given the stochastic nature of card distribution and the relatively small sample size, the reliability of the duplicate score rankings remains questionable.
2. The paper compares exclusively against general-purpose LLMs but omits comparisons with specialized Dou Dizhu AI systems like DouZero, which the authors cite as achieving superhuman performance through deep reinforcement learning. Without benchmarking against state-of-the-art specialized agents, it is impossible to assess whether the proposed approach truly achieves "expert-level" performance or merely represents the best performance within the limited LLM paradigm. This omission significantly weakens the claims about the model's strategic capabilities.
3. The evaluation framework entirely excludes human players of varying skill levels, making claims about "expert-level reasoning" unsubstantiated. The paper provides no empirical evidence comparing their best model (Qwen3-4B-GOFA with +17.25 duplicate score) against amateur, intermediate, or professional human Dou Dizhu players. This absence is particularly problematic given the authors' acknowledgment in Section 6 that their model's performance against "the broader, and at times deeply irrational, spectrum of strategies exhibited by human players remains to be further validated."
4. The ablation study only compares three coarse-grained configurations (baseline, victorious data, full GOFA) without isolating the individual contributions of the two core mechanisms: globally optimal decision alignment and real-time feedback augmentation. The paper provides no experiments showing the impact of using only one mechanism versus the other, making it impossible to determine which component is more critical or whether both are necessary. This limits understanding of the framework's key drivers and prevents principled refinement of the methodology.
5. The validation set comprises only 1,000 samples (~2.4% of the 41,884 GOFA samples), which may be insufficient to reliably guide model selection and early stopping for such a complex strategic task. Additionally, the paper lacks critical training details such as the number of training epochs, convergence curves, overfitting analysis, and the specific checkpoint selection criteria beyond "highest decision accuracy on validation set." The claim of training for "approximately one epoch" is vague and raises concerns about reproducibility and whether the model was adequately trained.

---

### Official Review · Reviewer_rUtg · 2025-10-29

**Soundness:** 2
**Presentation:** 3
**Contribution:** 2
**Rating:** 2
**Confidence:** 4

**Summary:**

This paper tests LLM strategy in Dou Dizhu using a duplicate tournament, GOFA data (God’s-eye check + in-game feedback), and a simple curriculum to boost Qwen3-4B. They claim better decision error rate and stronger robustness than baselines, but there exists flaws in design and conclusion.

**Strengths:**

1. Focusing on imperfect-information games fills a gap. Most prior LLM reasoning research targets perfect-information tasks or simple NLP tasks, while dynamic multi-agent imperfect-information scenarios are more representative of real-world strategic decision-making.
2. The "duplicate round-robin" format is a thoughtful attempt to measure skill rather than luck, which aligns with the need for fair LLM evaluation in stochastic games.
3. Fine-tuning a small 4B-parameter model instead of relying on large, costly LLMs like GPT-5 addresses a practical constraint relevant for deploying strategic AI agents.

**Weaknesses:**

1. The paper’s curriculum assumes "easy-to-hard" learning is optimal, but lacks an ablation control group. Without this control, it is impossible to verify:  Whether skipping the first stage leads to convergence failure;  or whether direct GOFA fine-tuning is better.
2. The duplicate score lacks statistical significance analysis. For example, the paper reports a score jump from -65.80 to 17.25 on Qwen3-4B-GOFA, but nostandard deviation. No analysis of round-to-round stability.
3. The construction way of validation set leads to a self-referential bias. The model’s decision accuracy is defined as alignment with GOFA annotations, but training and validation data share the same distribution. This only measures how well the model fits the training data, not how well it generalizes to unseen scenarios.
4. The "real-time in-game feedback" relies on LLMs to score decisions, but no calibration or consistency checks are performed: no disclosure of whether the scoring LLMs were trained on human-annotated good/bad decision samples; no cross-LLM consistency analysis; no clarity on scoring dimensions.
5. The God’s-eye view validation is fundamentally misaligned with real Dou Dizhu gameplay, also no quantification of this mismatch.
6. The paper claims the data framework "can be extended to other imperfect-information games"but provides no evidence, since the key mechanisms are Dou Dizhu-specific.
7. The paper does not explain how to extract core decision tokens from LLM outputs:
- Is structured output enforced via prompts? Or is keyword matching used?
- No quantification of how extraction methods affect loss calculation

**Questions:**

1. Will you add an ablation group that fine-tunes Qwen3-4B directly on GOFA data?
2. Can you provide 95% confidence intervals,standard deviation, and p-values for the duplicate scores of all models?
3. Will you collect an independent human expert validation set  and report the model’s accuracy on this set?
4. Can you disclose (1) how scoring LLMs were calibrated; (2) cross-LLM consistency; (3) specific scoring dimensions?
5. What is the overlap between "God’s-eye optimal decisions" and "top human players’ decisions under local information"? If overlap is low, how will you adjust the GOFA framework to align with real imperfect-information gameplay?
6. Can you conduct a small-scale experiment?
7. Can you detail the method used to extract decision tokens from LLM outputs? How different extraction methods affect model performance?

---

### Official Review · Reviewer_X6Dt · 2025-11-01

**Soundness:** 2
**Presentation:** 2
**Contribution:** 1
**Rating:** 2
**Confidence:** 4

**Summary:**

In this paper, the authors investigates this challenge using the popular card game Dou Dizhu as a representative testbed, aiming to enhance the reasoning and decision-making abilities of LLMs in such complex scenarios. First, the authors establish a rigorous and fair benchmark using a duplicate round-robin tournament to comprehensively evaluate the performance of several state-of-the-art LLMs. This evaluation provided a clear performance baseline and revealed that while these top-tier models are powerful, their significant computational cost motivates the development of smaller, more efficient alternatives. Furthermore, the authors propose a novel data
construction framework designed to bridge the information gap. Its core consists of two unique data curation mechanisms tailored for such games: globally optimal decision alignment via symmetric information and real-time in-game feedback augmentation.

**Strengths:**

First, the authors establish a rigorous and fair benchmark using a duplicate round-robin tournament to comprehensively evaluate the performance of several state-of-the-art LLMs. This evaluation provided a clear performance baseline and revealed that while these top-tier models are powerful, their significant computational cost motivates the development of smaller, more efficient alternatives.

Second, the authors propose a novel data construction framework designed to bridge the information gap. Its core consists of two unique data curation mechanisms tailored for such games: globally optimal decision alignment via symmetric information and real-time in-game feedback augmentation.

**Weaknesses:**

1. Using Dou Di Zhu to evaluate LLMs has some flaws: Only 3 LLMs can be involved in a single round, therefore, comparing more than 3 LLMs is difficult
2. Some related papers are not cited, such as Empowering LLMs in Decision Games through Algorithmic Data Synthesis, LLM-Based Explicit Models of Opponents for Multi-Agent Games, uno arena for evaluating sequential decision-making capability of large language models.

**Questions:**

NA

---

### Meta-Review · Area_Chair_Cvsq · 2025-12-24

**Summary:**

The paper proposes a framework for evaluating and enhancing Large Language Models (LLMs) in the imperfect-information game of Dou Dizhu. The authors introduce a "duplicate round-robin" benchmark to reduce variance and a data construction method (GOFA) involving "God's-eye" validation and simulated feedback.
However, the reviewers were unanimous in their recommendation to reject the paper, with all four reviewers assigning a score of 2 (Reject). The primary concerns driving this decision are:

1.	Multiple reviewers noted the critical omission of specialized Reinforcement Learning baselines (e.g., DouZero) or human expert comparisons. Without these, claims of "expert-level" performance are unsubstantiated.

2.	The evaluation, based on only 200 unique deals, was deemed insufficient for a high-variance card game. The lack of confidence intervals, standard deviations, or significance tests undermines the validity of the results.

3.	The "Duplicate Score" is a relative metric dependent on the opponent pool, making cross-paper or absolute performance comparisons impossible. Additionally, the metric was not formally mathematically defined.

4.	Concerns were raised regarding the validation set construction (self-referential bias), the lack of calibration for the LLM-based feedback mechanism, and ablation studies that failed to isolate the specific contributions of the proposed data curation mechanisms.

**Reviewer Concerns:**

Addressed by Rebuttal: None. The authors did not submit a rebuttal to address the reviewer inquiries or criticisms.

**Reviewer Scores:**

Given that the authors did not participate in the rebuttal phase, it is highly probable that the reviewers would maintain their original scores of 2.

The reviewers identified fundamental flaws in the experimental design—specifically the absence of standard RL baselines and statistical significance testing—that would require new experiments to resolve. Without a rebuttal or new data to refute these points, the reviewers' initial assessments regarding the paper's lack of readiness for publication stand firm.

---

### Decision · Program_Chairs · 2026-01-26

Reject